# Approximate Nullspace Augmented Finetuning for Robust Vision Transformers

Haoyang Liu[1†], Aditya Singh[2†], Yijiang Li[3], Haohan Wang[1]
[1]University of Illinois at Urbana-Champaign
[2]Sorted technologies
[3]University of California, San Diego
{hl57, haohanw}@illinois.edu
aditya.singh@sortedtech.io
yijiangli@ucsd.edu

Enhancing the robustness of deep learning models, particularly in the realm of vision transformers (ViTs), is crucial for their real-world deployment. In this work, we provide a finetuning approach to enhance the robustness of vision transformers inspired by the concept of nullspace from linear algebra. Our investigation centers on whether a vision transformer can exhibit resilience to input variations akin to the nullspace property in linear mappings, which would imply that perturbations sampled from this nullspace do not influence the model's output when added to the input. We start from the observation that many existing ViTs satisfy this property because their patch embedding layer has a non-trivial nullspace. Then, we extend the notion of nullspace to nonlinear settings and demonstrate that it is possible to synthesize approximate nullspace elements for ViT's encoder blocks through optimization. Finally, we propose a finetuning strategy for ViTs wherein we augment the training data with synthesized approximate nullspace noise. We find that our finetuning approach significantly improves the models' robustness to both adversarial and natural image perturbations.[1]

## 1. Introduction

The field of computer vision has experienced significant advances, marked by the emergence of Vision Transformers (ViTs) [1] as a notable milestone. Following this advancement, a series of architectural refinements have been explored [2–4], paving the way for the development of vision foundation models [5, 6] through the scaling up of both the model and dataset. Despite these strides, robustness continues to be a pivotal concern for their practical deployment, as they exhibit fragility in the face of imperceptible (adversarial) and perceptible perturbations.

Adversarial samples are generated by adding imperceptible noises to the input, aiming to cause the model to produce incorrect and overly confident predictions [7–9]. Perceptible perturbations are artifacts that arise from various operations, such as JPEG compression, simulated weather effects (fog, snow), or adjustments to the image's brightness, hue, or contrast, to name a few [10]. The semantic content of the image however, remains unchanged after perceptible or imperceptible perturbations. Hence, we expect the model to output similar predictions for perturbed and unperturbed images.

Applying transformations to the input during training, known as data augmentation, is one of the widely employed techniques for improving robustness. The underlying goal of applying augmentations is to enforce invariance (i.e., consistency) under a predefined set of perturbations. To induce adversarial robustness, worst-case adversarial perturbations are first identified through an optimization procedure and then used to train the model [11, 12]. For robustness against perceptible noise, augmentation strategies have evolved from simple transformations such as horizontal flips and rotations to more complex augmentations like MixUp [13], CutMix [14], and AugMix [15].

---

[1]Code is available at: https://github.com/Liu-Hy/ns-vit

Second Conference on Parsimony and Learning (CPAL 2025).

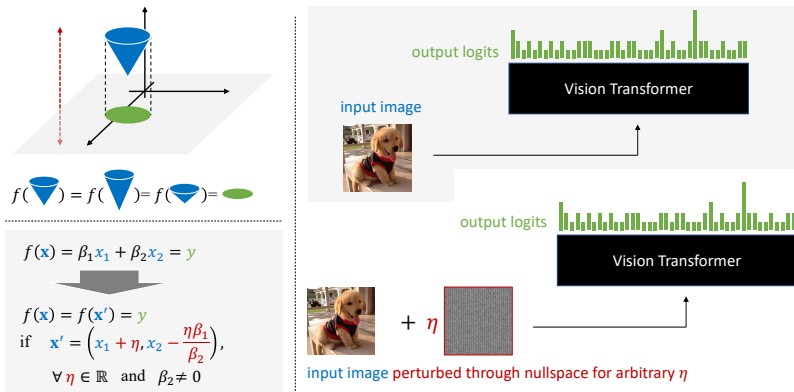

Figure 1: **An illustration of the nullspace in three cases (projection function, left top; linear function, left bottom; vision transformer, right)**. For these three cases, there exists some nullspace, such that the output of the function with respect to the input will remain unperturbed regardless of the perturbation strength. Also, the nullspace is function-specific (model-specific) and will not vary for different samples.

There is an observable divide in the treatment of these two types of robustness [16]. Adversarial noises are generated via an optimization process, whereas augmentations are defined heuristically often by domain experts. It has also been observed that standard data augmentation strategies, in isolation, do not improve adversarial robustness [17–19]. Additionally, adversarial training (training with adversarial perturbations) often leads to a drop in performance on non-adversarial images [11, 12, 20].

Here, we consider robustness a property of the model and thus agnostic to the noise type. To this end, we consider the nullspace as the central theme of our study. The nullspace, a fundamental concept in linear algebra, refers to the subspace of a domain that is mapped to zero by a linear mapping. By definition, any vector from the nullspace, when added to the input of the linear mapping, does not alter its output. In Figure 1, we present the concept of nullspace from different perspectives.

This paper first identifies that most off-the-shelf pre-trained ViT models exhibit a nontrivial nullspace due to the linear patch embedding layer. Since this layer is the first block of a ViT, any invariance to it implies invariance to the entire model. Consequently, a nontrivial nullspace also exists for ViTs. To further explore robustness, we define the approximate nullspace of the transformer encoder and use optimization methods to synthesize noise vectors approximating nullspace properties for nonlinear blocks. Finally, we propose fine-tuning the model using these synthesized nullspace-like elements as additive training data augmentation. This approach enlarges the approximate nullspace, enhancing the model's robustness. The main contributions of our paper include:

- We demonstrate connections between the robustness of vision transformers to the algebraic notion of nullspace, substantiated by experimental results showing that enlarging the approximate nullspace effectively improves the model robustness.
- We conduct comprehensive analysis on the existence of nullspace within transformer models. We establish the existence of nullspace at the patch embedding layer, and empirically identify an approximate nullspace at the nonlinear encoder level of transformers by validating their algebraic properties.
- We propose an effective data augmentation method by exploiting and enlarging the model's approximate nullspace, which enhances model robustness without architectural modifications and only involves fine-tuning with minimal additional data. Our method is empirically validated across multiple benchmark datasets, showing significant robustness improvements against adversarial and out-of-distribution scenarios.

## 2. Related Work

**Data augmentation and Invariance:** Data augmentation enforces invariance by training models to predict consistently across different input views, offering a theoretical improvement in estimating statistical risk [21, 22]. However, incorrect augmentation choices can degrade performance [21–23]. Early image augmentations such as flipping, cropping and rotation have evolved into advanced techniques such as MixUp [13], CutMix [14], and strategies for chain augmentations, including AutoAugment for policy optimization, TrivialAugment [24], and RandAug [25]. AugMix [15] combines transformations with a consistency loss, while differentiable augmentations optimize transformations for specific tasks [26, 27]. Hounie et al. [28] frame data augmentation as an invariance-constrained learning problem, using a relaxed invariance notion to model augmentation distributions. Unlike these approaches, our work avoids reliance on pre-defined augmentations.

**Robustness in ViTs:** Research highlights Vision Transformers (ViTs) as more robust than Convolutional Neural Networks (CNNs)[29, 30], with adversarial examples that exhibit low transferability between these architectures[31], although some studies offer counterpoints [32]. ViTs demonstrate insensitivity to patch-based transformations that distort semantics, relying on robust but nonindicative features [33]. Robustness-enhancing methods for transformer-based models are often model-agnostic, using data augmentation [34–37] and regularization [36, 38, 39], consistent with broader robustness frameworks [40, 41]. For example, Xiao et al. [34] masks image patches using class activation maps and refills them with random samples, while Chen et al. [38] adopts sharpness-aware optimization for a smoother loss landscape. However, these approaches focus on external modifications or optimization, often neglecting the intrinsic properties of the model.

**Nullspace and Neural Networks:** The study of nullspaces in neural networks began with Goggin et al. [42], who explored MLPs' universal approximation by comparing input nullspaces and outputs. Using the *learning XOR* example, they demonstrated that hidden layers enable MLPs to map inputs to targets even if the targets reside in the nullspace of the inputs. More recently, Sonoda et al. [43] mathematically analyzed nullspaces in fully connected networks.

In applications, Wang et al. [44] leveraged nullspaces in continual learning to map new tasks to the nullspace of existing ones. As a novel architecture, NullSpaceNet [45] mapped inputs from the same category to a joint nullspace rather than a feature space.

## 3. Nullspace and Invariance

When a mapping $f : \mathcal{X} \rightarrow \mathcal{Y}$ is invariant to some additive noise $\mathbf{v}$, it implies the following:

$$f(\mathbf{x} + \mathbf{v}) = f(\mathbf{x}) \quad \forall \mathbf{x} \in \mathcal{X}. \tag{1}$$

This invariance has interesting connections to the concept of *nullspace* in linear algebra. Formally, the nullspace of a linear mapping $f$ is a set $\mathcal{N}$ identified by $\mathcal{N} = \{\mathbf{v} \in \mathcal{X} | f(\mathbf{v}) = 0\}$. For a non-trivial nullspace $\mathcal{N} \neq \phi$, we have $f(\mathbf{x} + \mathbf{v}) = f(\mathbf{x}), \forall \mathbf{v} \in \mathcal{N}, \forall \mathbf{x} \in \mathcal{X}$. We can interpret this by saying that the linear mapping is invariant to the noise vector sampled from its nullspace. For brevity, we refer to this noise vector as **nullspace noise**.

### 3.1. Non-trivial Nullspace of the Patch Embedding Layer

Vision transformer [1] is a function $f_\omega$ with $\omega$ as the trainable weights. The function takes as input an image $\mathbf{x} \in \mathcal{X}^{c \times h \times w}$ and outputs a classification response $\mathbf{y} \in \mathcal{Y}^k$ over $k$ categories. $c$ is the number of channels (typically 3 for red, green, and blue), $h, w$ correspond to height and width of the input image. This neural network function can be broken down into 3 stages, namely:

- *patch embedding stage*, $f_\theta : \mathcal{X}^{c \times r \times r} \rightarrow \mathcal{U}^d$. This steps projects the input image patch of predetermined dimensions $c$, $r$ and $r$ to a one-dimensional embedding of length $d$. It is ensured that patches have no overlaps between them. Hence, the number of such non-overlapping patches generated from the input image are $m = \frac{h \times w}{r^2}$.

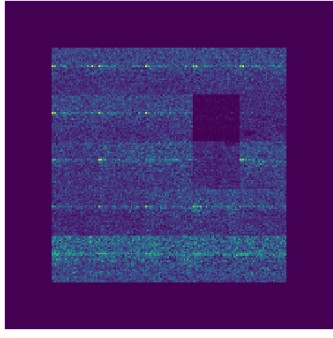
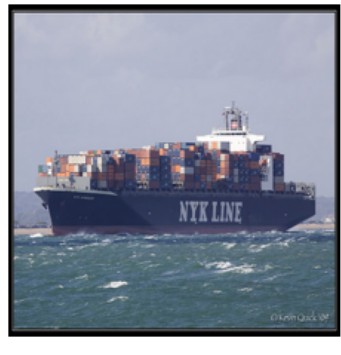
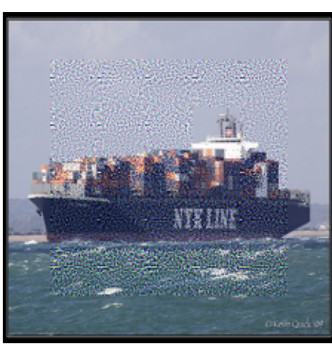

(a) Sample null-space noise      (b) Clean input image      (c) Noisy input image

Figure 2: **An example of nullspace noise.** We show (a) sample input image, (b) noise generated by the basis vectors of the nullspace and (c) noisy image as a result of adding the nullspace noise to the input. Model's predictions for the clean and noisy inputs are identical.

- *self-attention stage*, $f_\phi : \mathcal{U}^{(m+1)\times d} \to \mathcal{V}^{(m+1)\times d}$. In the next step, the generated patch embeddings are passed through layers of self-attention modules to process long range interactions amongst them. Apart from the $m$ patch embeddings an additional embedding in form of `cls` token is utilised in this step.
- *classification stage*, $f_\psi : \mathcal{V}^d \to \mathcal{Y}^k$. The final step is to perform the $k$-way classification. For this, we simply keep the processed encoding corresponding to `cls` token and project it through a linear classification layer.

Since the first layer of the ViT is a linear mapping, according to the rank-nullity theorem, it always has a non-trivial nullspace if $cr^2 > d$. In practice, for many ViT-based architectures, we find that this is the case. In Table 1, we report the identified nullspace dimensions for off-the-shelf pre-trained ViT models.

Given the weights of the patch embedding layer $f_\theta$, finding its nullspace is a standard practice [46–48]. Let $B_\theta = \{\mathbf{b}_1, \mathbf{b}_2, \dots \mathbf{b}_k\}$ be the $k$ basis vectors for this nullspace, we can sample an element from $\mathcal{N}_\theta$ as:

Table 1: **Nullspace dimensions for pre-trained ViT models.** Nullspace is trivial ($\mathbf{0}$) when embedding dimension exceeds input dimension.

| Model | Patch Size | Emb. Dim. | Null Dim. |
|---|---|---|---|
| tiny | $16 \times 16$ | 192 | 576 |
| small | $32 \times 32$ | 384 | 2688 |
| | $16 \times 16$ | 384 | 384 |
| base | $32 \times 32$ | 768 | 2304 |
| | $16 \times 16$ | 768 | 2 |
| | $8 \times 8$ | 768 | 0 |
| large | $32 \times 32$ | 1024 | 2048 |
| | $16 \times 16$ | 1024 | 0 |

$$\mathbf{v} = \lambda_1 \mathbf{b}_1 + \lambda_2 \mathbf{b}_2 + \cdots + \lambda_d \mathbf{b}_k. \tag{2}$$

The property of such a sample will be that the output of the patch embedding will effectively remain preserved, $f_\theta(\mathbf{x}+\mathbf{v}) = f_\theta(\mathbf{x})$. Since the output after the first layer remains unaffected, the final output of the classification remains unchanged. In Figure 2, we provide visualization of noise synthesized using basis vectors. This noise can be added to *any* input image with complete invariance. In Section 3.2, we explore if it possible to learn a nullspace-like counterpart for the non-linear blocks of ViTs.

## 3.2. The Generalized Nullspace from the Encoder

So far we have demonstrated that a non-trivial nullspace exists for the patch embedding layer, and hence the entire vision transformer is invariant to all perturbations in that space. We move further down the structure of ViT and investigate whether the encoder is also invariant to certain perturbations. The self-attention layer is non-linear, which means the notion of nullspace cannot be directly applied to $f_\phi$. However, the *invariance* property that can be implied from the nullspace of linear functions, that any vector from this set will not alter the function's output when added to any input, is still desirable in the nonlinear case when it comes to the robustness of neural models.

In fact, data augmentation can often be formulated as a process of adding noise to the input and enforcing invariance. Therefore, to study the ViTs' inherent invariance to input perturbations, we extend the notion of nullspace to the nonlinear setting and define the *Generalized Nullspace*, $\tilde{\mathcal{N}}_\phi$, of the transformer encoder $f_\phi$, as below:

$$\tilde{\mathcal{N}}_\phi = \{\mathbf{v}|f_\phi(\mathbf{u} + \mathbf{v}) = f(\mathbf{u}) \quad \forall \mathbf{u} \in \mathcal{U}\}, \tag{3}$$

Here, we use the tilde accent $\tilde{\phantom{x}}$ to distinguish $\tilde{\mathcal{N}}_\phi$ from the conventional nullspace $\mathcal{N}_\phi$. We term it the Generalized Nullspace because it depicts invariance in both linear and nonlinear settings, and that for a linear fuction $f_\theta$ we have $\mathcal{N}_\theta \subseteq \tilde{\mathcal{N}}_\theta$, since any vector sampled from the conventional nullspace of a linear function satisfies this invariance property. If such a set exists, it directly implies that the transformer model is robust to certain perturbations in the input space. Our theoretical analysis established the following sufficient conditions for the existence of a nontrivial generalized nullspace. (The complete proof is given in Appendix A.)

**Proposition 1.** *Consider a self-attention layer with $h$ heads and $\{(\mathbf{Q}_i, \mathbf{K}_i, \mathbf{V}_i)\}_{i=1}^h$ as its query, key and value projection matrices. If the following conditions are met*

1. *$\mathbf{Q}_i\mathbf{K}_i^\top$ is symmetric for $i = 1, \dots, h$*
2. *The row space $\mathrm{R}(\mathbf{V}_i^\top) \subseteq \mathrm{R}(\mathbf{Q}_i\mathbf{K}_i^\top)$ for $i = 1, \dots, h$*
3. *for some $m \neq n$, $\mathbf{Q}_m\mathbf{K}_m^\top$ has colinearity with $\mathbf{Q}_n\mathbf{K}_n^\top$, i.e. for some $k$ the $k$th row of $\mathbf{Q}_m\mathbf{K}_m^\top$, denoted as $\mathbf{r}_{m,k}$, satisfies $\mathbf{r}_{m,k} \neq \mathbf{0}$ and $\mathbf{r}_{m,k} \in \mathrm{R}(\mathbf{Q}_n\mathbf{K}_n^\top)$*

*then there exists at least one $\mathbf{W}$ such that $\mathbf{W} \neq \mathbf{0}$ and $\mathrm{head}_i(\mathbf{X} + \mathbf{W}) = \mathrm{head}_i(\mathbf{X})$ for all attention head $i$ in this layer and arbitrary $\mathbf{X}$.*

*Remark* 1. Condition 1 can be met if $\mathbf{Q}_i$ and $\mathbf{K}_i$ satisfy some special relation. For example, let $\mathbf{PDP}^{-1}$ be a diagonalization of a real symmetric matrix $\mathbf{A}$. If $\mathbf{Q}_i = \mathbf{BP}$ and $\mathbf{K}_i = \mathbf{B}(\mathbf{P}^{-1})^\top\mathbf{D}$, then we have $\mathbf{Q}_i\mathbf{K}_i^\top = \mathbf{BAB}^\top$ to be symmetric.

In addition, evidence has shown that, $\mathbf{Q}_i\mathbf{K}_i^\top$ can be empirically symmetric, especially for ViTs, when the attention heads are visualized and correlation of parameters is calculated [49].

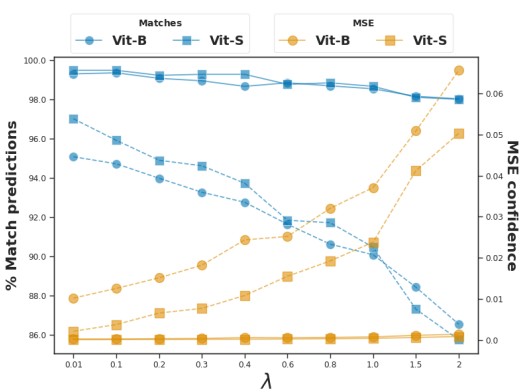 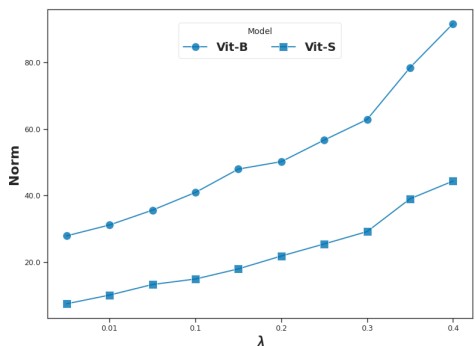

(a) Noise influence on the model output under different regularization strengths

(b) $\ell_2$ norm of learned noise under different regularization strengths

Figure 3: **Exploratory experiments on the generalized nullspace.** (a) Solid lines ($-$) represents the model performance under the learned noise, and dashed lines ($\cdots$) represent the performance after random permutation of the elements of the learned noise vector. (b) by changing the regularization strengths, we explore noise in the generalized nullspace at different magnitudes.

## 3.3. Synthesizing (approximate) nullspace noise

Although our theory suggests a sufficient condition for the existence of generalized nullspace, analytically finding $\tilde{\mathcal{N}}_\phi$ or probing its existence for generic transformers is challenging. Thus, as an

exploratory experiment, we employ a numeric method: we search for individual element, $\tilde{\mathbf{v}}_\phi$, of this set. This element is an additive perturbation that brings minimal influence to the output of $f_\phi$ on the data distribution. We introduce a regularization term on the norm of $\tilde{\mathbf{v}}_\phi$ to prevent the trivial solution of $\mathbf{0}$.

$$\mathcal{L}_\phi(\tilde{\mathbf{v}}) = \underbrace{\mathbb{E}_{\mathbf{u} \in \mathcal{D}} \left\| f_\psi(f_\phi^0(\mathbf{u} + \tilde{\mathbf{v}})) - f_\psi(f_\phi^0(\mathbf{u})) \right\|}_{\mathfrak{I}} - \lambda \log(\|\tilde{\mathbf{v}}\|). \tag{4}$$

Here, $\|\cdot\|$ is the $\ell_2$ norm, $f_\phi^0$ is the representation of the `cls` token output by $f_\phi$, and $\lambda$ is the regularization coefficient. $\mathfrak{I}$ resembles a weaker notion of invariance compared to Equation (1). Equation (4) minimizes the $\ell_2$ norm between the predicted logits with and without the noise. Alongside the self-attention stage, we have also incorporated the classification stage into the loss, since the target of our study is to minimize the impact on the final output of the network. To learn the noise vector, we initialize $\tilde{\mathbf{v}}$ by sampling from a uniform distribution, and minimize the loss with gradient descent. We use ViT-S and ViT-B models with patch size 32 for evaluation. We employ ImageNette [50] as the dataset for this experiment, which is a subset of ImageNet consisting of 10 categories. We learn $\tilde{\mathbf{v}}$ on the training dataset ($\approx 9500$ images) and perform evaluation on the validation set ($\approx 4000$ images).

To quantitatively evaluate learned $\tilde{\mathbf{v}}_\phi$, in Figure 3 (a) we report the percentage of matching classifications with and without learned nullspace noise, and the mean squared error computed at the output probabilities (hereafter "MSE confidence"). We consider a prediction to be matched if the assigned category for input is the same with and without adding the perturbation. By varying the regularization strength, we get noise vectors of different magnitude (Figure 3 (b)), all being fairly benign to the model's predictions. However, if we randomly reset the vectors' direction by permuting their elements, the noise significant degrades the model's predictions.

The experiment shows the feasibility of learning elements that approximately conform to our definition of generalized nullspace with good precision, and also indicates that at different magnitudes there are certain directions in the input space toward which the perturbation is fairly benign to the model. In Appendix E, we further empirically show that the learned noise vectors exhibit good properties under scalar multiplication and convex combinations within certain scope of parameters, similar to the closure property of a vector space.

## 4. Nullspace Noise Augmented Finetuning

In this section, we investigate the application of the synthesized nullspace noise. As we discussed previously, the model is weakly invariant to the learnt noise ($\mathfrak{I}$ in Equation (4)) and the set as a result of this relaxed notion is an approximate nullspace. To more accurately quantify this, we define the $\epsilon$-*Approximate Generalized Nullspace* as follows (later called "$\epsilon$-approximate nullspace" or "approximate nullspace" for brevity):

$$\tilde{\mathcal{N}}_\phi(\epsilon) = \{\tilde{\mathbf{v}} | \mathbb{E}_{\mathbf{u} \in \mathcal{D}} \left\| f(\mathbf{u} + \tilde{\mathbf{v}}) - f(\mathbf{u}) \right\| \le \epsilon\}, \tag{5}$$

where $f(\cdot) = \text{Softmax}(f_\psi(f_\phi^0(\cdot)))$. It is easy to verify that $\forall \epsilon > 0, \mathbf{0} \in \tilde{\mathcal{N}}_\phi(\epsilon)$, and that $\forall \epsilon_2 > \epsilon_1 > 0$, $\tilde{\mathcal{N}}_\phi(\epsilon_1) \subseteq \tilde{\mathcal{N}}_\phi(\epsilon_2)$.

We believe that the existence of approximate noise vectors is a property of the model. As these vectors exhibit relaxed invariance, we also believe that they play a key role in model's inherent robustness under a variety of distribution shifts. Hence, if we can further improve invariance on approximate nullspace elements, we can potentially make the model more robust. With this belief, **we propose to fine-tune a pre-trained ViT with the learnt nullspace noise vector as an added (encoder level) input perturbation.** The motivation behind this is to enlarge the (approximate nullspace) set of noise vectors to which the model is invariant.

Formally, we employ a bi-level optimization approach, where the inner problem finds the best noise vector and the outer problem finds the model that is the most tolerant to such noise, as shown below.

$$\min_{\phi} \quad \mathbb{E}_{\mathbf{u} \in \mathcal{D}} \, \ell(f_\psi(f_\phi^0(\mathbf{u} + \tilde{\mathbf{v}}_\phi^*)), \mathbf{y})$$
$$\text{where} \quad \tilde{\mathbf{v}}_\phi^* = \arg\max_{\tilde{\mathbf{v}}} \|\tilde{\mathbf{v}}\| \quad \text{s.t.} \quad \tilde{\mathbf{v}} \in \mathcal{N}_\phi(\epsilon). \tag{6}$$

Here, $\ell(\cdot)$ is the cross-entropy loss. While this optimization problem can also be solved in different ways, we use an efficient heuristic: we initialize the noise with a large enough sampling limit, minimize $\mathcal{L}_\phi(\tilde{\mathbf{v}})$ by gradient descent according to the loss function in Equation 7, and early stop it as soon as it enters $\mathcal{N}_\phi(\epsilon)$, as shown in Equation 8.

$$\mathcal{L}_\phi(\tilde{\mathbf{v}}) = \mathbb{E}_{\mathbf{u} \in \mathcal{D}} \, \|f_\psi(f_\phi^0(\mathbf{u} + \tilde{\mathbf{v}})) - f_\psi^0(f_\phi(\mathbf{u}))\| \tag{7}$$
$$\hat{\mathbf{v}}^* = \text{SGD}(\mathcal{L}_\phi(\tilde{\mathbf{v}}), \tilde{\mathbf{v}}_0, \epsilon). \tag{8}$$

Here, $\hat{\mathbf{v}}_\phi^*$ is the approximate solution for $\tilde{\mathbf{v}}_\phi^*$, $\text{SGD}(\mathcal{L}_\phi(\tilde{\mathbf{v}}), \tilde{\mathbf{v}}_0, \epsilon)$ denotes the gradient descent algorithm that minimizes the loss $\mathcal{L}_\phi(\tilde{\mathbf{v}})$ starting from its initial value $\tilde{\mathbf{v}}_0$ until it satisfies the condition $\mathcal{L}_\phi(\tilde{\mathbf{v}}) < \epsilon$. The noise norm starts from a large value and gets gradually reduced during the process. When early stopping is triggered, we obtain noise vectors that are close to the boundary of the $\epsilon$-approximate nullspace. For more details of our method, please refer to Algorithm 1 in Appendix B.

## 5. Experiments

### 5.1. Implementation Details

In this section, we conduct evaluation of our nullspace augmented finetuning method (Section 4) on several benchmarks. By making the model more tolerant to noise in the $\epsilon$-approximate nullspace, we hope to expand the nullspace itself and observe its effect on the model's robustness under different settings.

Starting from a pretrained model, we use the $\epsilon$-approximate nullspace noise as data augmentation to fine-tune the model. The noise is generated every 40 training steps according to Equation (8) with an $\epsilon$ of 0.03. The experiment was done within one epoch of training on the ImageNet-1k [51] dataset. We used the vanilla `ViT-small` and `ViT-base` models, and `ViT-base(DAT)` which is the current SOTA on ImageNet-C dataset on the EasyRobust benchmark[2], trained using Discrete Adversarial Training [52]. We evaluated the model performance in a wide range of settings to test its performance on the i.i.d dataset, under adversarial attacks and distribution shifts. For adversarial attacks we utilize FGSM [7], DamageNet [53], PatchFool [54] and CW [55]. Among them, FGSM and CW are gradient-based white-box attacks, DamageNet consists of pre-generated adversarial examples, and PatchFool targets localized, adversarial patches of an image. For distribution shift we employ ImageNet-C [10], ImageNet-A [56], ImageNet-V2 [57], Imagenet-R [58], ImageNet-Sketch [59] and Stylized-Imagenet [60]. ImageNet-C consists of validation images modified by applying corruptions in the form of weather effects, noises, etc. ImageNet-A applies the imagenet objects in hard contexts. ImageNet-R and ImageNet-Sketch consist of imagenet categories in different art forms. ImageNet-Stylized applies texture transfer onto the ImageNet validation images to create shape-texture contradictions.

We use the EasyRobust library [61] for code implementation and the checkpoints of `ViT-base(DAT)`. For more implementation details please see our supplementary document.

### 5.2. Experiment: Robustness Evaluation

We evaluated the effect of nullspace finetuning to improve the robustness of vision transformers under different settings. We used the official mCE score as the evaluation metric for ImageNet-C, where a lower mCE indicates better robustness, and we used the accuracy score for all other settings. We used $100 - \text{mCE}$ before taking the average in all settings.

---

[2]https://github.com/alibaba/easyrobust

Table 2: **Effect of our nullspace augmented finetuning (NS) method on different models evaluated on multiple benchmark datasets.** Excluding DAT, vanilla ViT-S and ViT-B, the values for the baselines are directly reported from the corresponding papers. For DAT, we report the reproduced results following their evaluation setting.

| Methods | Clean | Adversarial Robustness | | | | Out of Distribution Robustness | | | | | | Average |
| --- | --- | --- | --- | --- | --- | --- | --- | --- | --- | --- | --- | --- |
| | | PatchFool | CW | FGSM | DamageNet | A | C↓ | V2 | R | Sketch | Stylized | |
| ViT-S | 74.19 | 0.68 | 4.63 | 13.79 | 29.82 | 16.35 | 71.13 | 62.51 | 34.67 | 14.26 | 12.15 | 26.54 |
| ViT-S + NS (ours) | **77.47** | **19.10** | **9.37** | **25.95** | **32.43** | **20.77** | **55.98** | **66.5** | **41.61** | **25.67** | **16.02** | **34.45** |
| ViT-B | 77.68 | 15.92 | 12.54 | 25.65 | 38.69 | 23.88 | 62.16 | 66.05 | 41.63 | 16.31 | 17.97 | 34.01 |
| ViT-B + MixUp [13] | 77.80 | – | – | – | – | 12.20 | 61.80 | – | 34.90 | – | – | – |
| ViT-B + RandAugment [25] | 79.10 | – | – | – | – | – | 43.60 | – | 23.00 | – | – | – |
| ViT-B + PR [33] | 78.20 | – | – | – | – | – | 47.60 | – | 21.40 | – | – | – |
| ViT-B + RandAugment + PR | 79.30 | – | – | – | – | – | 43.60 | – | 23.80 | – | – | – |
| ViT-B + AugMix [15] | 78.80 | – | – | – | – | – | 42.20 | – | 24.90 | – | – | – |
| ViT-B + AugMix + PR | 79.30 | – | – | – | – | – | **41.60** | – | 25.70 | – | – | – |
| ViT-B + SAM [38] | 79.90 | – | – | – | – | – | 43.50 | 67.50 | 26.40 | – | – | – |
| RobustViT-B [62] | 80.40 | – | – | – | – | 23.00 | – | 69.80 | 35.40 | **35.80** | – | – |
| ViT-B + NS | **81.42** | **23.52** | **14.23** | **36.50** | **40.44** | **24.55** | 47.82 | **70.25** | **44.85** | 26.35 | **19.02** | **39.39** |
| ViT-B + DAT[52] | 81.47 | 22.64 | 23.59 | 48.80 | 43.31 | 23.83 | 45.95 | **70.24** | **48.68** | 36.94 | **23.99** | 43.41 |
| ViT-B + DAT + NS | 81.33 | **24.14** | **23.61** | **48.98** | **43.67** | **24.22** | **45.91** | 70.14 | 48.48 | **37.25** | 23.87 | **43.61** |

Table 3: **Comparison of our NS method with PGD-based adversarial robustness methods of Madry and TRADES.** We report the performance for a ViT-S model.

| Method | clean | FGSM | DamageNet | A | C (↓) | V2 | R | Sketch | Stylized |
| --- | --- | --- | --- | --- | --- | --- | --- | --- | --- |
| ViT-S | 74.19 | 13.79 | 29.82 | 16.35 | 71.13 | 62.51 | 34.67 | 14.26 | 12.15 |
| Madry | 70.53 | **39.37** | **49.91** | 9.37 | 81.74 | 58.88 | 39.04 | 21.36 | 10.76 |
| TRADES | 74.02 | 38.85 | 36.28 | 16.53 | 73.11 | 63.37 | 40.86 | **26.43** | 13.22 |
| NS | **77.47** | 25.95 | 32.43 | **20.77** | **55.98** | **66.5** | **41.61** | 25.67 | **16.02** |

The result in Table 2 shows that our nullspace finetuning method consistently improves the robustness of models under distribution shifts and adversarial attacks, yielding a large gain in average performance for the vanilla `ViT-small` and `ViT-base model`, and slightly outperforms various baselines consistently while also slightly outperforming DAT. This not only shows that our nullspace finetuning method is effective but also validates our previous hypothesis about the connection between the tolerance to nullspace and the robustness of transformer models.

## 5.3. Experiment: Adversarial Finetuning

In this experiment, we compare our method with fine-tuning using two PGD adversarial training methods, Madry [63] and TRADES [12] on the ViT-S model. TRADES, in each training iteration, generates adversarial examples using PGD and updates the model's parameters to minimize the worst-case loss on these adversarial examples while also minimizing the standard classification loss on clean data. Madry, on the other hand, focuses exclusively on minimizing the worst-case loss on adversarial examples. In Table 3, we observe that Madry and TRADES provide better performance for adversarial evaluation. This is expected as the methods are catered for improving adversarial robustness. However, this exclusivity leads to relatively poorer performance in a wider benchmark evaluation. Compared to our method, Madry and TRADES perform considerably lower in the natural OOD setting.

## 5.4. Enlarged Approximate Nullspace

To gain more insight about the dynamics of our nullspace finetuning method, we monitor the $l_2$ norm of the learned noise and various performance metrics during the training, as shown in Fig. 4. Before the nullspace finetuning, it was hard to optimize the noise into the $\epsilon$ region even with increased training, so the norm started with a high value. As the training starts, we find that the noise was always able to enter the $\epsilon$ region. In Appendix C, we show the MSE probability of the learned noise vectors after each round of noise learning, which were all smaller than $\epsilon$. More importantly, the norm

Table 4: **Impact of $\epsilon$ on the final performance**. Moreover, we also compare our NS method against random $\epsilon$ noise based finetuning.

| $\epsilon$ | Finetuning | FGSM | DamageNet | A | C ($\downarrow$) | V2 | R | Sketch | Stylized |
|------------|------------|------|-----------|------|------|-------|-------|--------|----------|
| 0.01 | NS | **26.04** | **33.65** | **20.45** | 56.26 | **66.47** | **41.4** | **23.34** | **15.85** |
| | Random | 21.54 | 28.81 | 17.07 | **55.13** | 61.98 | 34.97 | 14.43 | 12.14 |
| 0.03 | NS | **25.95** | **32.43** | **20.77** | 55.98 | **66.5** | **41.61** | **25.67** | **16.02** |
| | Random | 23.18 | 29.61 | 16.91 | **54.68** | 62.2 | 35.05 | 14.77 | 12.34 |
| 0.1 | NS | **25.38** | **33.09** | **20.16** | 56.41 | **66.47** | **40.42** | **22.66** | **15.78** |
| | Random | 23.93 | 30.56 | 16.47 | **54.52** | 62.48 | 34.66 | 14.99 | 12.35 |

of the learned noise gradually increases along the process of model fine-tuning. The fluctuation may have mainly resulted from the randomness in mini-batches and the optimization dynamics.

The model allows for noises with larger and larger norms to be within $\epsilon$-approximate, which informally suggests an enlarging $\epsilon$-approximate nullspace. Accompanied by the trend is the increase in robustness scores in both OOD and adversarial settings, which corroborates our findings.

### 5.5. Ablation Study

We conduct an extensive study to analyse the performance of our method under choice of $\epsilon$. Furthermore, we also compare our approach with a simple baseline of using an $\epsilon$ noise sampled from a Gaussian distribution.

From Table 4, we can infer that the nullspace noise based finetuning is relatively robust to the choice of $\epsilon$. Moreover, compared to using randomly generated $\epsilon$-noise, our nullspace based training provides significant performance boost. This observation stands across different values of $\epsilon$.

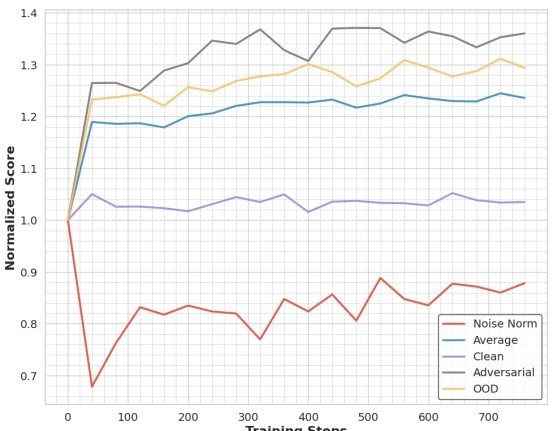

Figure 4: **Change trend of multiple metrics with training steps.** "Adversarial" is the average performance of the 4 adv. robustness settings, "OOD" is the average score on the six OOD datasets, and "avg" is the total average. All values are divided by their initial values to show the trend more clearly.

## 6. Discussion

**Applications in Model Patenting** In addition to the applications we discussed, we consider another potential usage of our findings is to patent a ViT after a model is trained, as the nullspace will be unique property of any set of weights of certain ViT architectures. Different from the existing line of research in model watermarking [64–66], the patenting through nullspace will not require any additional steps during training, although will face limited usage scenarios in comparison.

**Applications in Image Watermarking** Using the nullspace noise, it is possible to apply signatures onto input images without harming the output or operability of the networks. In the supplementary document, we present the cases where certain marks in form of nullspace noise can be superimposed on any desired input image.

**Potential Limitation about the Nullspace Approximation** Different from the nullspace defined in linear algebra, the nullspace of the entire ViT can only be approximated because of the non-linearity in the network architecture. However, it is worthy mentioning that we can still calculate the exact nullspace of ViT if we only consider the patch embedding layer, through which, our results will qualitatively deliver the same message, but with quantitative differences.

# 7. Conclusion

In this work, we have explored the concept of nullspace in Vision Transformers (ViTs) to understand their robustness. Our findings demonstrate that a non-trivial nullspace indeed exists for Vision Transformers, a direct consequence of the patch embedding layer. This discovery implies that there are elements that, when added to an input, do not affect the output of the network, potentially offering an explanation for the robustness exhibited by ViTs. Moreover, we have extended the definition of nullspace, preserving a property that implies invariance of a mapping's output to input perturbations, and empirically identified a space that exhibits such property within the input space of the non-linear transformer encoder. By linking the presence of nullspace with our extended definition to the general robustness of a network, we were able to devise a new approach to improve the robustness of ViTs. Our empirical results suggest that fine-tuning ViTs with the learnt nullspace noise can significantly enhance their robustness to a variety of robustness benchmarks.

This study offers a new perspective to the robustness of vision transformers. We believe these findings can assist in furthering the robustness of ViTs, potentially facilitating advancements in the development of more resilient models. Looking forward, there is more to explore in this direction. Future research could focus on the development of efficient algorithms for learning nullspace and investigate its presence in other architectures and layers of deep neural networks.

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

# Supplementary Material

## A. Proof of Proposition 1

Let $d$ be the hidden dimension of the attention layer. $\mathbf{Q}_i, \mathbf{K}_i \in \mathbb{R}^{d \times d_k}$ where $d_k = d/h$. $\mathrm{rank}(\mathbf{Q}_i \mathbf{K}_i^\top) \leq \mathrm{rank}(\mathbf{K}_i^\top) \leq d_k$. Consider the sum of row spaces $S = \mathrm{R}(\mathbf{Q}_1 \mathbf{K}_1^\top) + \mathrm{R}(\mathbf{Q}_2 \mathbf{K}_2^\top) + \cdots + \mathrm{R}(\mathbf{Q}_h \mathbf{K}_h^\top)$. $S$ is a subspace of $\mathbb{R}^d$. For $i = 1, \ldots, h$, choose a basis for $\mathrm{R}(\mathbf{Q}_i \mathbf{K}_i^\top)$, denoted as $B_i = \{\mathbf{b}_1, \cdots, \mathbf{b}_{n_i}\}$, $|B_i| = n_i \leq d_k$. Without loss of generality, let $\mathbf{r}_{m,k} \in B_m$.

$S = \mathrm{span}(\bigcup\limits_{i=1}^{h} B_i)$, so

$$\dim(S) = \dim\left(\mathrm{span}\left(\bigcup_{i=1}^{h} B_i\right)\right) = \dim\left(\mathrm{span}\left(\left(\bigcup_{\substack{i=1 \\ i \neq m}}^{h} B_i\right) \cup (B_m \setminus \{\mathbf{r}_{m,k}\})\right)\right)$$

$$\leq \left|\left(\bigcup_{\substack{i=1 \\ i \neq m}}^{h} B_i\right) \cup (B_m \setminus \{\mathbf{r}_{m,k}\})\right| \leq (h-1)d_k + (d_k - 1) = d - 1. \tag{9}$$

So, $\exists \mathbf{w} \in \mathbb{R}^d, \mathbf{w} \neq \mathbf{0}$ and $\mathbf{w} \in S^\perp$. This means for $i = 1, \ldots, h$, $\mathbf{w} \in \left(\mathrm{R}\left(\mathbf{Q}_i \mathbf{K}_i^\top\right)\right)^\perp$, $\mathbf{w} \in \mathrm{N}\left(\mathbf{Q}_i \mathbf{K}_i^\top\right)$. By condition 2, $\mathrm{N}(\mathbf{V}_i) \supseteq \mathrm{N}(\mathbf{Q}_i \mathbf{K}_i^\top)$, so $\mathbf{w} \in \mathrm{N}\left(\mathbf{Q}_i \mathbf{K}_i^\top\right) \cap \mathrm{N}(\mathbf{V}_i^\top)$.

Then, we can choose $\mathbf{W}$ wherein each row is a multiple of $\mathbf{w}$. We have $\mathbf{W}\mathbf{V}_i = \mathbf{0}$, and for any input to the encoder $\mathbf{X} \in \mathbb{R}^{n \times d}$,

$$\mathbf{W}\mathbf{Q}_i \mathbf{K}_i^\top \mathbf{X}^\top + \mathbf{X}\mathbf{Q}_i \mathbf{K}_i^\top \mathbf{W}^\top + \mathbf{W}\mathbf{Q}_i \mathbf{K}_i^\top \mathbf{W}^\top = \mathbf{0}. \tag{10}$$

Consider the output of attention head,

$$\begin{aligned}
\mathrm{head}_i(\mathbf{X} + \mathbf{W}) &= \mathrm{Softmax}\left(\frac{(\mathbf{X} + \mathbf{W})\,\mathbf{Q}_i \mathbf{K}_i^\top (\mathbf{X} + \mathbf{W})^\top}{\sqrt{d_k}}\right)(\mathbf{X} + \mathbf{W})\mathbf{V}_i \\
&= \mathrm{Softmax}\left(\frac{\mathbf{X}\mathbf{Q}_i \mathbf{K}_i^\top \mathbf{X}^\top + \mathbf{W}\mathbf{Q}_i \mathbf{K}_i^\top \mathbf{X}^\top + \mathbf{X}\mathbf{Q}_i \mathbf{K}_i^\top \mathbf{W}^\top + \mathbf{W}\mathbf{Q}_i \mathbf{K}_i^\top \mathbf{W}^\top}{\sqrt{d_k}}\right)\mathbf{X}\mathbf{V}_i \\
&= \mathrm{Softmax}\left(\frac{\mathbf{X}\mathbf{Q}_i \mathbf{K}_i^\top \mathbf{X}^\top}{\sqrt{d_k}}\right)\mathbf{X}\mathbf{V}_i = \mathrm{head}_i(\mathbf{X}).
\end{aligned} \tag{11}$$

Adding the noise $\mathbf{W}$ does not change the output of any attention head for arbitrary input $\mathbf{X}$, which completes our proof.

# B. Algorithm and implementation details

We present the algorithm of our data augmentation with nullspace noise in Algorithm 1.

---

**Algorithm 1:** Data augmentation with nullspace noise

---

1   **Input:** transformer model with patch embedding layer $f_e$, encoder $f_\phi$ and linear classifier $f_\psi$ parameterized by $e, \phi, \psi$ respectively; training data $\mathcal{T}$; batch size $B$; sampling limit $A$; noise nullity threshold $\epsilon$; noise learning rate $\eta_v$; model learning rate $\eta_f$; number of outer iterations $K$, noise training step $T$, model training step $S$

2   **for** $k = 0, \cdots, K - 1$ **do**

3      Sample initial noise $\mathbf{v} \sim \mathrm{U}(-\mathrm{lim}, \mathrm{lim})$

4      **for** $t = 0, \cdots, T - 1$ **do**

5         Sample a minibatch $(\mathbf{X}, \mathbf{y}) \sim \mathcal{T}$

6         Compute $\mathbf{U} \leftarrow f_e(\mathbf{X})$

7         Compute logits $\mathbf{Z} \leftarrow f_\psi(f_\phi^0(\mathbf{U})), \mathbf{Z}' \leftarrow f_\psi(f_\phi^0(\mathbf{U} + [\mathbf{v}]))$   `# "[v]" means broadcasting the noise v along the sample dimension`

8         Compute $\delta \leftarrow \frac{1}{B} \sum_{i=1}^{B} \|\mathrm{Softmax}(\mathbf{z}_i') - \mathrm{Softmax}(\mathbf{z}_i)\|^2$   `# zi is sample logit`

9         **if** $\sigma < \epsilon$ **then**

10            **break**

11         **end**

12         Calculate $\ell \leftarrow \frac{1}{B} \sum_{i=1}^{B} \|\mathbf{z}_i' - \mathbf{z}_i\|^2$

13         Update $\mathbf{v} \leftarrow \mathbf{v} - \nabla_{\mathbf{v}} \ell$

14      **end**

15      **for** $s = 0, \cdots, S - 1$ **do**

16         Sample a minibatch $(\mathbf{X}, \mathbf{y}) \sim \mathcal{T}$

17         Compute $\mathbf{U} \leftarrow f_e(\mathbf{X})$

18         Compute logits $\mathbf{Z} \leftarrow f_\psi(f_\phi^0(\mathbf{U})), \mathbf{Z}' \leftarrow f_\psi(f_\phi^0(\mathbf{U} + [\mathbf{v}]))$

19         Compute loss $\mathcal{L} \leftarrow \frac{1}{B} \sum_{i=1}^{B} (\ell(\mathbf{z}_i, y_i) + \ell(\mathbf{z}_i', y_i))$, where $\ell$ is the cross-entropy loss

20         Update model parameters $(\psi, \phi, e) \leftarrow (\psi, \phi, e) - \nabla_{(\psi, \phi, e)} \mathcal{L}$

21      **end**

22   **end**

23   **Output: model weight** $(\psi, \phi, e)$

---

**Hyperparameters** We fine-tuned the ViT model for $K = 20$ rounds in all settings. In each round, we initialized the noise with sampling limit $A = 3$, optimized it with learning rate $\eta_v = 0.1$ and set a threshold of $\epsilon = 0.03$. We empirically found that $T = 3000$ is enough to trigger early stopping so that the learned noise satisfies the $\epsilon$ threshold. We used $\eta_f = 10^{-5}$ to fine-tune the model for $S = 40$ iterations in each round. We set batch size $B = 128$, and slightly different from the vanilla SGD in Alg 1, we used the AdamW optimizer [67] and cosine learning rate scheduler with defualt hyperparameters for both the noise and the model training.

The original `ViT-B + DAT` model [52] used the Exponential Moving Average (EMA) for evaluation[3], so we also used EMA to evaluate the performance of `ViT-B + DAT` fine-tuned with our method. For all the other settings, we used single model without ensemble for evaluation. We used $\epsilon = 1/255$ for the FGSM attack consistent with [52].

**Computation time** The experiments were conducted on a combination of A100, V100 GPUs and a 3090 GPU, depending on the availability. Although we only used about 10% of the ImageNet-1k [51] training data to fine-tune the model, the main computation time is on training the nullspace noise. One run of our experiment (20 rounds) takes the time roughly equivalent to 8 epochs of standard training on ImageNet-1k.

---

[3] https://github.com/alibaba/easyrobust

## C. Change trend of the noise influence with the fine-tuning steps

Beside the trend of noise norm and performance metrics in Fig. 4, we also keep track of the influence of the learned noise in terms of MSE probability (3.2) at every 80 steps of the model fine-tuning. As shown in Table 5, the noise influence is always lower than $\epsilon = 0.03$, which means early stopping is triggered and the model enters the $\epsilon$ region.

Table 5: MSE probability of the noise at different fine-tuning steps.

| Fine-Tuning Step | 40 | 120 | 160 | 280 | 360 | 440 | 520 | 600 | 680 | 760 |
|---|---|---|---|---|---|---|---|---|---|---|
| MSE Probability | 0.028 | 0.027 | 0.026 | 0.029 | 0.028 | 0.029 | 0.027 | 0.025 | 0.028 | 0.026 |

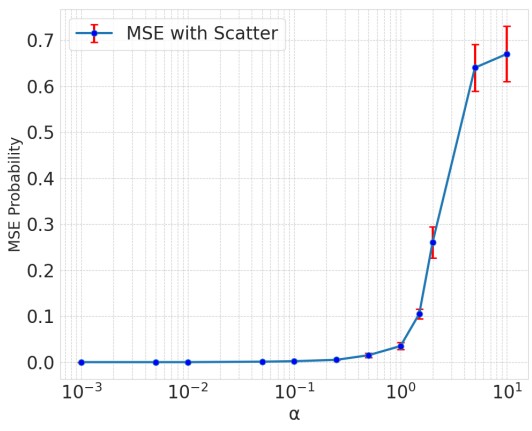

(a) Influence of $\epsilon$ noise under multiplication with different $\alpha$

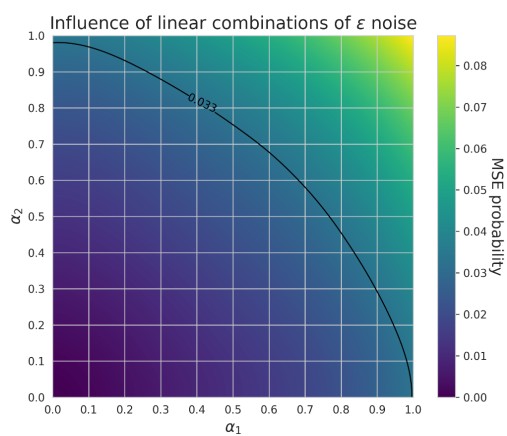

(b) Influence of $\epsilon$ noise under convex combination with different $\alpha_1, \alpha_2$

Figure 5: Validation of the properties of the $\epsilon$-approximate nullspace.

## D. Approximate Nullspace Properties

To explore the property of the $\epsilon$-approximate nullspace, we conduct an experiment to observe the behavior of the learned noise vectors under scalar multiplication and convex combination. For this, we first construct a set of $m$ $\epsilon$-approximate nullspace vectors $\mathbf{V} = \{\mathbf{v}_i\}_{i=1}^m$ starting from different random initializations using $\epsilon = 0.033$. For scalar multiplication, we vary the scaling factor $\alpha$ and report the mean influence of $\alpha\mathbf{v}$ on the model's predictions in terms of MSE probabilities (Figure 5(a)). For convex combination, we sample $n$ different pairs of nullspace vectors from $\mathbf{V}$, denoted as $\mathbf{P} = \{(\mathbf{v}_{\mathcal{J}_{k,1}}, \mathbf{v}_{\mathcal{J}_{k,2}})\}_{k=1}^n$, where $\mathcal{J}_{k,1}, \mathcal{J}_{k,2} \in \{1, 2, \ldots, m\}, \forall k \in \{1, \ldots n\}$. Then, we vary $\alpha_1$ and $\alpha_2$ between $[0, 1]$ with a grid size of $0.1$, and for each combination of $(\alpha_1, \alpha_2)$, we evaluate the influence of the convex combination $\alpha_1\mathbf{v}_{\mathcal{J}_{k,1}} + \alpha_2\mathbf{v}_{\mathcal{J}_{k,2}}$ on the model's prediction in MSE probability, averaged over all values of $k$. In practice we set $m = 100, n = 10$. The influence of the linear combined noise at each point of the grid is visualized as a heatmap as shown in Figure 5(b).

The results in Figure 5 show that the approximate nullspace has similar property to vector space in terms of closure under addition and scalar multiplication within a certain range of coefficients. When the scaling factor $\alpha < 1$, we see a clear trend that the MSE probability of the scaled noise is less than $\alpha\epsilon$. In the linear combination case, the line $\alpha_1 + \alpha_2 = 1$ is well within the contour line of MSE probability being 0.033, showing that the convex combination of a pair of $\epsilon$ noise vector is still $\epsilon$-approximate.

# E. Watermarking Images

Watermarking as image, usually used to convey ownership information or verify content of the data, has been studied extensively [68–71]. A watermark can be either imperceptible or perceptible. and perceptible watermarking applies a noticeable marker to convey the protected nature of the data [72]. In this section, we explore to utilize nullspace noise to apply a perceptible watermark on the image which does not alter the test-time forward process.

Figure 6 shows an example watermarking approach using the nullspace noise. Here, we emboss the ICML logo onto the natural images. The resulting modified image, attains the final predictions close to the original image. ($100\%$ match in the final output prediction and $10^{-4}$ difference in the predicted confidence value of the assigned class.)

**Method details:** With basis vectors of the nullspace, we can construct a watermark to be overlaid on the original image without affecting the output of the network. Given a source (user's image) and a target image (watermark), we simply need to estimate the scalar parameters corresponding to the basis vectors to satisfy $\sum_{i=0}^{i<m} \mathbf{e}_i \lambda_i = \mathbf{v}_\theta \approx \Delta \mathbf{x}_j$.

$\mathbf{e}_i$ are the basis vectors for the nullspace, $\lambda_i$ are their corresponding scalar co-efficients which are to be determined and $\Delta \mathbf{x}_j$ is the changed required to convert $j^{\text{th}}$ original image patch to $j^{\text{th}}$ watermark image patch. This can be achieved through a constrained optimisation of the following form:

$$\min \|\Delta \mathbf{x}_j - \sum_{i=0}^{i<m} \mathbf{e}_i \lambda_i\|_p. \tag{12}$$

Here, $\Delta x_j$ is the difference between the $j^{th}$ channel of a source and target image and $\lambda_i$ is the $i^{th}$ nullspace basis vector of the patch embedding layer with the corresponding variable scalar $e_i$. We use a least-square solver to solve for the solution (Available readily with packages such as Numpy).

# F. Targeted Nullspace Noise

Due to the dimension reduction effect of the patch embedding layer in most ViTs, we can transfer an image to be visually similar to another image by human perception, without changing the output of the original image perceived by the model. This differs from adversarial examples in the following aspects:

1. The working direction to construct an adversarial example is the other way around. If the transformed image is to be considered an adversarial example, then our source becomes the target for adversarial training and our target becomes the source.

2. Generating targeted nullspace noise requires no backpropagation through the network

3. Not only does the final prediction on the transformed image matches the source image, the saliency maps also match. This is displayed in Fig. 7

Though the transformation is not perfect, we can spot that the transformed images are visually similar to target images rather than source images. Even with this difference in the input space, transformed images and source images are classified into the same category with roughly the same confidence.

As recent studies have shown, fooling can also be extended to the interpretability methods (XAI) Dombrowski et al. [73] partially due the limitations exposed by recent studies [73–75]. However, in contrast to these works aiming to fool specific XAI method, our nullspace noise only depends on the model, not the XAI method.

In Fig. 7(b), we show the interpretability maps as generated by LRP [76]. From the figure, we can observe that the heatmaps generated by source and transformed images are identical whereas, the

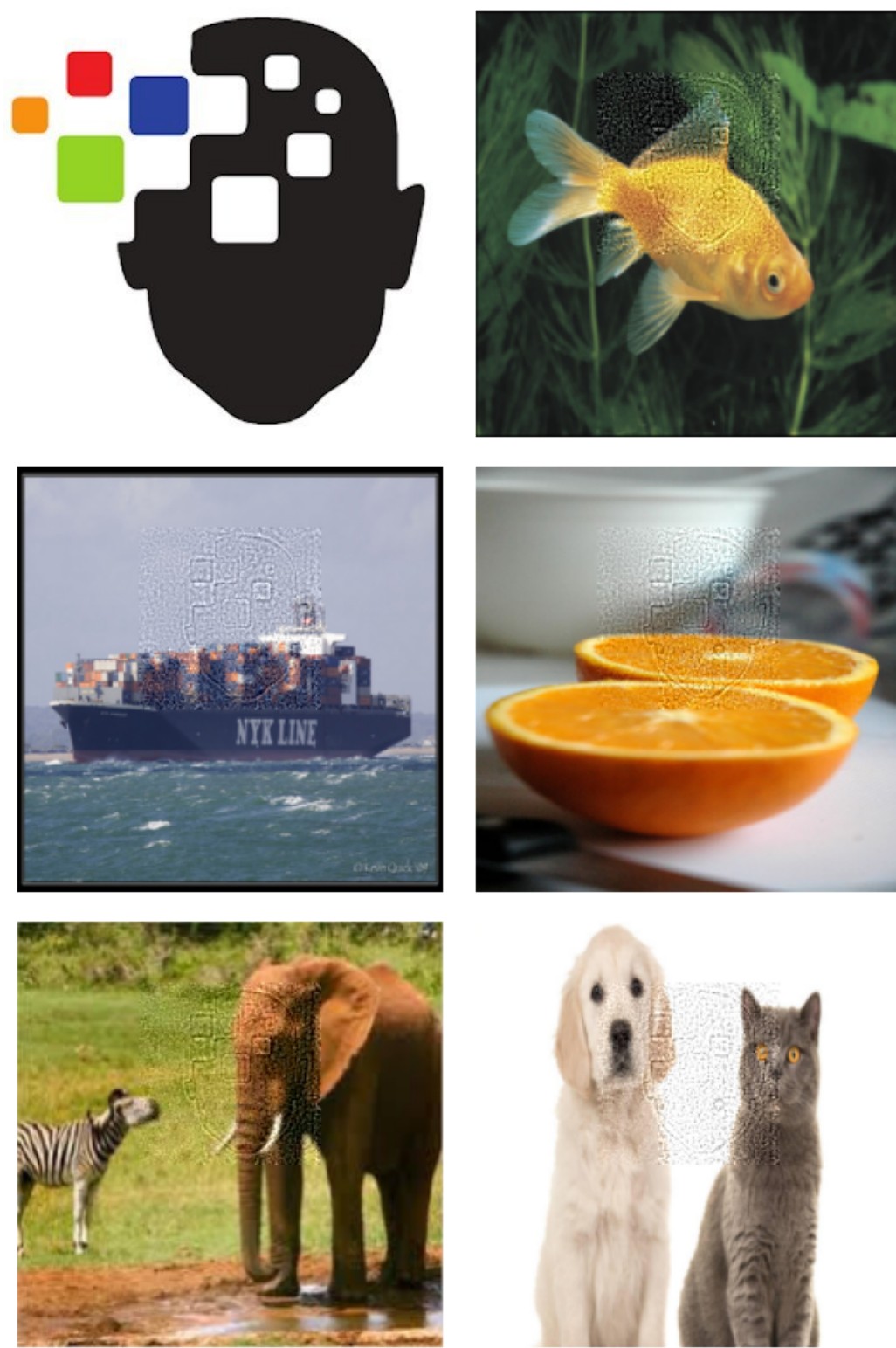

Figure 6: **Watermark superposition using the nullspace basis vectors.**

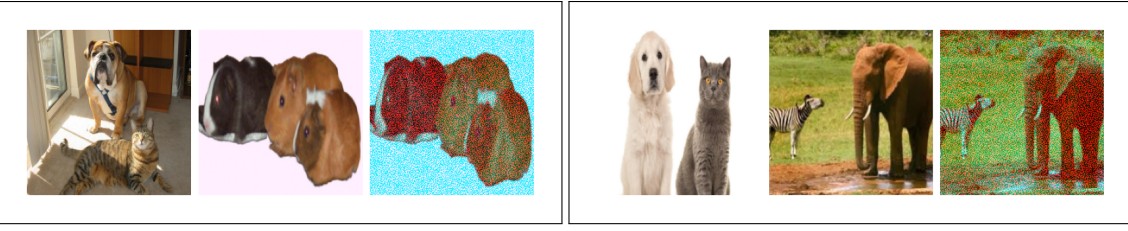

(a) Triplet of Source, target and transformed images

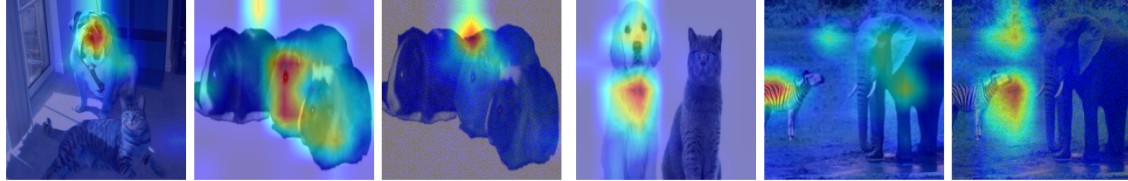

(b) Saliency maps for the corresponding images from the row above.

Figure 7: **Targeted nullspace noise.** Transformed images appear visually as target images but are interpreted as source images by the model. The equivalence between source and transformed images is not only in terms of the final predictions but also in the interpretability maps depicted in (b).

transformed image heatmaps substantially differ from target images'. Though only reported for LRP, we observed that a similar observation holds across different interpretability approaches. Here, we only presented the results on LRP, as in the context of ViTs, we found the heatmaps from other methods to be lacking (also pointed out by authors of LRP).

In Fig. 8 we show the saliency maps generated by different XAI methods. Even though the maps generated by methods other than LRP are poor (hard to interpret), we see that the source and transformed respond similarly to these methods.

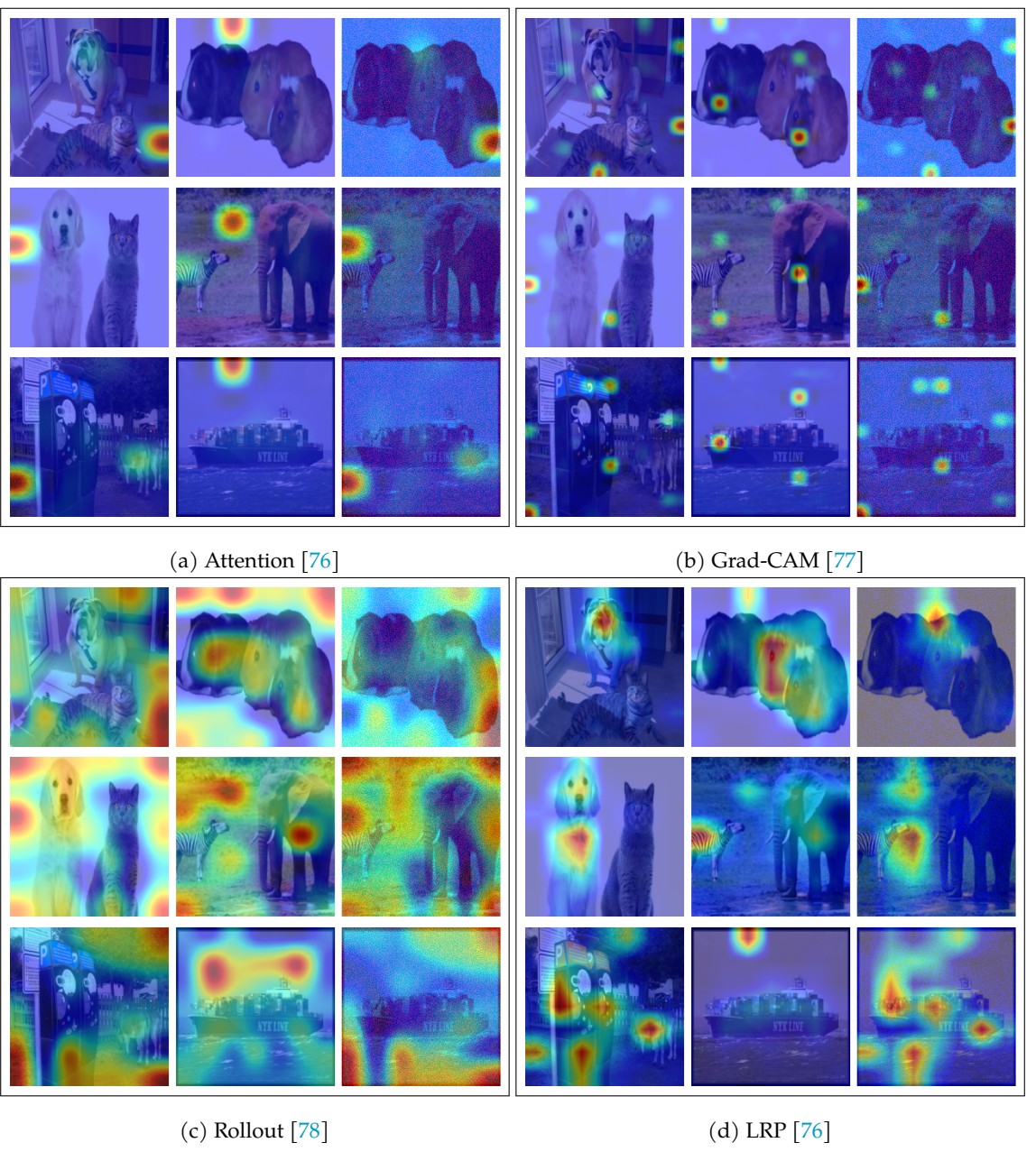

(a) Attention [76]

(b) Grad-CAM [77]

(c) Rollout [78]

(d) LRP [76]

Figure 8: **Interpretability maps generated via different methods for (source, target, transformed) images**

