# OpenReview forum: "Approximate Nullspace Augmented Finetuning for Robust Vision Transformers"
_CPAL.cc/2025/Proceedings_Track — CPAL 2025 (Proceedings Track) Oral_

### Official Review · Reviewer_H3Wu · 2025-01-07
**review for submission 77**

**Rating:** 7
**Confidence:** 3

**Review:**

This paper presents a novel approach to enhancing the robustness of vision transformers (ViTs) by leveraging the algebraic concept of nullspace. The study extends the idea of nullspace from linear to nonlinear settings and synthesizes approximate nullspace elements to augment the training process. This augmentation significantly improves the robustness of ViTs against both adversarial and natural image perturbations.

[Strengths]:
1. Innovative Conceptual Foundation: The approach of using nullspace properties from linear algebra to enhance the robustness of nonlinear models like ViTs is highly innovative and shows a creative integration of classical mathematical theories with modern deep learning architectures.
2. Empirical Validation: The paper provides extensive empirical evidence to demonstrate the effectiveness of the proposed finetuning strategy. It showcases significant improvements in model robustness across various benchmark datasets, which is a strong testament to the practical value of the research.
3. Detailed Theoretical Analysis: The paper offers a thorough theoretical exploration of the nullspace concept within ViTs, including proofs and propositions that enrich the understanding of why and how the method works.

[Drawbacks]:
1. Potential Overfitting to Synthesized Noise: While the method shows improvements in robustness, there is a potential risk that the model could become overfitted to the synthesized nullspace noise, which may not generalize well to entirely new types of perturbations or real-world scenarios.
2. Computational Resource Intensity: The synthesis of nullspace elements and the required optimizations might demand significant computational resources, which could limit the applicability of the approach in resource-constrained environments.

---

### Official Review · Reviewer_oNXW · 2025-01-10
**Some parts felt unclear, impractical, but nonetheless interesting work**

**Rating:** 6
**Confidence:** 3

**Review:**

This paper studies improving robustness for vision transformers. The authors (1) show that ViTs have nontrivial invariances in both the input patch embedding and self-attention layers, (2) show that approximate null space noise can be recovered through optimization, and (3) use this null space noise for data augmentation.


Strengths:
- I found the premise of the paper -- reinforcing approximate null spaces as a tool for improving robustness interesting to think about.
- The theoretical analysis seemed reasonable.
- The practical applications discussion (such as model patenting and image watermarking) is useful motivation.

Weaknesses/questions:
- A fundamental choice I felt was underdiscussed is: why do we perturb along the null space in order to expand the null space?
	- For example, why not perturb in directions orthogonal to the null space?
	- Both choices seem like they could be argued for from different directions, more discussion would be nice.
- The method appears impractical for larger models or most real-world use cases. The computational overhead of the bi-level optimization problem appears very high. The epsilon hyperparameter also appears to need to be tuned on a per-problem basis.
- Some presentation, like Figure 4, could use improvement.

Overall:
I found the paper's core ideas of approximate null spaces being inevitable in existing vision transformer architectures and the fact that they can be used for data augmentation interesting. While the method leaves some room for future improvement, the paper is well-executed and insightful. I would vote to accept.

---

### Official Review · Reviewer_bNn1 · 2025-01-12
**Review of Submission 77**

**Rating:** 6
**Confidence:** 3

**Review:**

This work investigate a new type of input noise of vision transformers, defined by the nullspace of model. Based on the nullspace noise, they propose a new fine-tuning method can enhance the generalization of the proposed methods.

(+) The concept is nullspace noise is interesting.

(+) Experiments contains plenty of ablation studies and models/datasets.

(+) The manucript is well writtern and easy to follow.

(-) Whether the proposed method can help enhance the natural noise (https://github.com/hendrycks/robustness)?

(-) The fine-tuning method seems like compatible with CNN-based method? Is there any discussion or experiment results to demonstrate this.

(-) Based on the definition of nullspace, f(x+v) = f(x), and f(v) = 0, Does it requies the function satisfying f(x+v) = f(x) + f(v) and why it's validate?

---

### Meta-Review · Area_Chair_b5gW · 2025-02-03

**Recommendation:** Accept (Oral)
**Confidence:** 4

**Metareview:**

This work investigates fine tuning strategies to enhance the robustness of vision transformers (ViTs). The key insight many existing ViTs architectures have non-trivial nullspace elements, both in an exact sense (due to the linear patch embedding layer) and an approximate sense (due to other effects in nonlinear layers). The authors propose fine-tuning ViTs by synthesizing nullspace-like elements which are then added to the training data as a data augmentation step. This effectively enlarges the model's nullspace along non-semantic dimensions, enhancing robustness.

Reviews were generally positive, finding the paper to be insightful and well-motivated. Some reviewers thought the idea of perturbing along the nullspace was underdeveloped. Additionally, there were concerns that the proposed bi-level optimization technique was too computational intensive. However, I feel the authors address these concerns adequately in their rebuttals.

Overall, this paper makes significant contributions towards understanding robustness in ViT. The authors give a comprehensive treatment, starting from the definition of non-linear nullspace vectors in ViTs, to theoretical guarantees of their existence, to practical algorithms for generating approximate null-space vectors, and finally an extensive empirical validation with the proposed data augmentation approach. I will also note that I found the paper to be very well-written. Therefore, my recommendation is to accept.

---

### Decision · Program_Chairs · 2025-02-11

Accept (Oral)